# Changes in m6A in Steatotic Liver Disease

**DOI:** 10.3390/genes14081653

**Published:** 2023-08-19

**Authors:** Belinda J. Petri, Matthew C. Cave, Carolyn M. Klinge

**Affiliations:** 1Department of Biochemistry, University of Louisville School of Medicine, Louisville, KY 40292, USA; belinda.petri@louisville.edu; 2Center for Integrative Environmental Health Sciences (CIEHS), University of Louisville, Louisville, KY 40292, USA; matt.cave@louisville.edu; 3Hepatobiology and Toxicology Center, University of Louisville, Louisville, KY 40292, USA; 4Division of Gastroenterology, Hepatology & Nutrition, Department of Medicine, University of Louisville School of Medicine, Louisville, KY 40292, USA

**Keywords:** m6A, RNA modifications, NAFLD, NASH, fatty liver, epitranscriptome

## Abstract

Fatty liver disease is one of the major causes of morbidity and mortality worldwide. Fatty liver includes non-alcoholic fatty liver disease (NAFLD) and non-alcoholic steatohepatitis (NASH), now replaced by a consensus group as metabolic dysfunction-associated steatotic liver disease (MASLD). While excess nutrition and obesity are major contributors to fatty liver, the underlying mechanisms remain largely unknown and therapeutic interventions are limited. Reversible chemical modifications in RNA are newly recognized critical regulators controlling post-transcriptional gene expression. Among these modifications, N6-methyladenosine (m6A) is the most abundant and regulates transcript abundance in fatty liver disease. Modulation of m6A by readers, writers, and erasers (RWE) impacts mRNA processing, translation, nuclear export, localization, and degradation. While many studies focus on m6A RWE expression in human liver pathologies, limitations of technology and bioinformatic methods to detect m6A present challenges in understanding the epitranscriptomic mechanisms driving fatty liver disease progression. In this review, we summarize the RWE of m6A and current methods of detecting m6A in specific genes associated with fatty liver disease.

## 1. Introduction

Chronic nutrient overload results in impaired glucose and lipid metabolism in the liver resulting in lipid accumulation in hepatocytes (steatosis) as the starting point for the spectrum of nonalcoholic fatty liver disease (NAFLD). The prevalence of NAFLD is increasing with 36–48% of the U.S. population affected [1,2,3]. NAFLD ranges from steatosis to non-alcoholic steatohepatitis (NASH), characterized by hepatocellular cell death, inflammation, and fibrosis with accumulation of proinflammatory monocyte-derived macrophages [4]; to cirrhosis and hepatocellular cancer (HCC) [1]. A group of national liver associations recently issued a consensus statement suggested replacing NAFLD with metabolic dysfunction-associated steatotic liver disease metabolic dysfunction-associated steatotic liver disease (MASLD) [5]. The rationale, in part, for the change is that MASALD was considered less stigmatizing in nomenclature for liver disease and MASALD incorporates at least one of five cardiometabolic disease risk factors [5]. For the purposes of this review, we will use the terms NAFLD and NASH due to their prevalence in the reported literature included herein. There are currently no approved therapeutic agents approved for treating NAFLD or NASH [6]. The farnesoid X receptor (FXR) agonist obeticholic acid recently failed to gain approval by the Food and Drug Administration for the treatment of NASH, given its side effects [7,8]. NAFLD is associated with obesity-related metabolic changes, i.e., insulin-resistance and type 2 diabetes mellitus (T2DM), and its prevalence is higher in men than premenopausal women [9,10]. However, after age 70, women have a higher prevalence of NAFLD than men [11]. Women treated with tamoxifen [2,12,13,14] or aromatase inhibitors [15] as endocrine therapies to prevent recurrence of estrogen receptor α positive (ERα+) breast cancer have a higher prevalence of NAFLD, notably in the context of obesity [16]. Studies in mice have demonstrated that estrogens have preventive metabolic effects against NAFLD mediated by hepatic estrogen receptor α (ERα) [17,18]. In addition to obesity, other contributors to NAFLD include genetic variants (SNPs) [19], epigenetics, e.g., altered DNA methylation and miRNA expression [20,21]; gut microbiome dysbiosis [1,2,22], and exposure to metabolism disrupting environmental chemicals (MDCs), e.g., polychlorinated biphenyls (PCBs) [23].

Another layer of coding events in regulatory control is the “epitranscriptome”, i.e., post-transcriptional chemical RNA modifications. Over 170 chemical modifications of transcribed mammalian RNA, including mRNA, rRNA, tRNA, and non-coding (nc) RNAs that regulate transcript fate and function have been identified [24,25,26]. These chemical ‘marks’ on RNA are dynamically regulated in response to cell stress and changes in cellular homeostasis [25,27,28]. N(6)-methyladenosine (m6A) is the most common dynamic modification of the mRNA transcriptome and regulates the transcription, processing, stability and the cellular location of mRNAs, long non-coding RNAs (lncRNAs), eRNAs, rRNA, tRNA, circRNAs, and primary microRNAs (pri-miRNAs) [29,30]. m6A is the best-studied epitranscriptome mark with 45 results in PubMed in a search for “m6A and fatty liver”. m6A is regulated by proteins called “readers, writers, and erasers” (RWE). The m6A modification occurs within the conserved RRACH (DRACH) motif (R = A, G; H = A, C, U) [31] which is recognized by the METTL3 writer (methyltransferase) complex [32,33]. Protein ‘readers’ of m6A regulate transcript fate by altering splicing (AS), subcellular location, stability, degradation, and translation [29,32,34].

There is one recent review on this topic that covered the role of immune cells in the liver and how m6A modification regulates immune cells and genes in lipid homeostasis in the liver as implicated by studies with overexpression of fat mass and obesity associated (FTO), a m6A ‘eraser’, in HepG2 cells [35]. Here we will review the state of knowledge about m6A and other RNA modifications in fatty liver disease.

## 2. N6-Methyladenosine (m6A)

Recent research has highlighted the important role that m6A modifications play in various cellular processes, including development [36,37], differentiation [38,39], and stress responses [40,41]. In Table 1, the key functions of m6A RNA metabolism are summarized to show how m6A modifications influence the fate of RNA. Dysregulation of m6A modification has been linked to a variety of diseases including cancer [42,43,44,45] and liver disease [46,47,48]. Therefore, there is significant interest in understanding the molecular mechanisms underlying m6A regulation and the potential for targeting this pathway for therapeutic purposes.

### m6A “Writers, Readers, and Erasers”

The intricate interplay involving three groups of key m6A modifiers, referred to as “readers, writers and erasers (RWE)“, plays a crucial role in shaping the outcomes of m6A modifications on RNA metabolism and cellular processes in both normal and stress conditions [58]. These effectors are responsible for the addition, removal, and interpretation of m6A modifications, respectively. Figure 1 provides an overview of RWE involved in m6A modifications in the context of NAFLD. The m6A methyltransferase complex (the m6A “writer” complex), is responsible for catalyzing the m6A methylation on RNA [59,60,61]. The m6A writer complex includes the catalytic methyltransferase like 3 (METTL3) which forms a stable complex with methyltransferase like 14 (METTL14) forming the m6A-METTL Complex (MAC) in which METTL3 transfer the methyl group from S-adenosyl methionine (SAM) to the adenosine residue within the RRACH consensus motif in RNA [62,63]. In addition to the MAC, the m6A writer complex includes regulatory subunits, i.e., Wilms’ tumor 1 (WT1)-associating protein (WTAP) [64], Vir like m6A methyltransferase associated (VIRMA) [65,66,67], RNA binding motif protein 15 (RBM15) [68], RNA binding motif protein 15B (RBM15B) [68,69], and zinc finger CCCH-type containing 13 (ZC3H13) [70]. These proteins interact with MAC and modulate its activity and/or target specificity [71,72,73].

m6A readers specifically bind to RNA molecules containing the m6A and mediate the downstream effects of the m6A modification on RNA metabolism, including mRNA stability [75,76], splicing [77,78], and translation [79,80]. There are several known families of m6A readers that contain specialized domains or motifs that allow them to specifically recognize and bind to m6A-modified RNA molecules [81,82]. The YTH domain family proteins (YTHDF1, YTHDF2, YTHDF3, YTHDC1, and YTHDC2) play a critical role in regulating mRNA decay and translation [83,84]. YTHDC1 and YTHDC2 are localized in the nucleus and regulate alternative splicing and mRNA export [85,86,87]. YTHDF1, YTHDF2, and YTHDF3 are localized in the cytoplasm and promote mRNA decay [75,88,89]. YTHDF2, for example, is an m6A reader that binds to m6A-modified mRNAs and recruits the CCR4-NOT deadenylase complex, leading to mRNA degradation [90]. In contrast, YTHDF1 and YTHDF3 are involved in enhancing mRNA translation by promoting the recruitment of translation initiation factors to the m6A-modified mRNA [80,91,92]. The heterogeneous nuclear ribonucleoprotein (HNRNP) family of m6A reader proteins regulate mRNA splicing and stability [93]. There are 16 HNRNP family members named alphabetically A-U, classified by their RNA binding domains and have diverse structural features and functions (reviewed in [94]). For example, HNRNPC bound to m6A-modified pre-mRNA and promoted exon skipping in activated T-cells [95]. HNRNPA2B1 (A2B1) stabilized m6A-modified AKT3 mRNA in multiple myeloma cells [96] and HNRNPA1 stabilized SIRT1 mRNA, delaying cellular senescence in HeLa cells [97]. HNRNPs have been shown to play important roles in several steps of miRNA biogenesis. A2B1 binds to pri-miRNAs and to DGCR in the DROSHA complex to promote their processing [98]. Insulin-like growth factor 2 mRNA-binding proteins (IGF2BP1, IGF2BP2, and IGF2BP3) are also m6A readers that regulate mRNA stability and translation [91]. IGF2BP2 is critical in regulating hepatic metabolism (reviewed in [99]).

m6A erasers are demethylases specifically remove the m6A modification from RNA molecules restoring the A to its unmodified RNA state [100]. There are currently two known m6A erasers for mRNA: the fat mass and obesity-associated protein (FTO) [100] and the alkB homolog 5 (ALKBH5) [101]. FTO was originally identified in genome-wide association studies (GWAS) as having a modest, but significant, contribution to childhood and adult obesity [102]. FTO was reported to share motifs with Fe(II)- and 2-oxoglutarate-dependent oxygenases and showed high mRNA expression in the hypothalamic nuclei and in the arcuate nucleus that regulates satiety and feeding behaviors [103]. FTO was later identified as an m6A demethylase in vitro and in HeLa and HEK-293T cell studies [100]. FTO regulates a variety of cellular processes, including adipogenesis [104], energy homeostasis [105], and response to stress [106]. FTO depletion in HepaRG and Huh-7 cells decreased CYP2C8 expression, demonstrating a role for post-transcriptional regulation of cytochrome P450s in drug-metabolism [107]. In a study investigating the effect of the phytoestrogen resveratrol on m6A methylation in high fat diet (HFD)-fed mice, resveratrol increased hepatic Fto transcript abundance that was downregulated with excess nutrition, thus decreasing hepatic m6A RNA methylation [108]. Fusaic acid, a mycotoxin found in contaminated foods, increased global m6A methylation in mouse livers by decreasing Fto expression [109].

There is evidence that ALKBH5 is the demethylase specificly responsible for removing the methyl group from m6A -modified mRNA, whereas FTO is the m6Am demethylase [110,111,112]. The molecular mechanisms by which m6A erasers remove the m6A modification are not yet fully understood, but it is thought that these enzymes use oxidative demethylation reactions to remove the methyl group from the N6 position of the adenosine [113,114]. Overall, the identification and characterization of m6A erasers have provided critical insights into the dynamic nature of the m6A modification and its role in regulating gene expression and cellular processes in liver.

## 3. Methods for the Detection of m6A Modifications

Driven by a combination of technological innovations and increased understanding of the functional importance of m6A modifications in RNA, the field of m6A detection methods has recently undergone significant advancement. These developments (reviewed in [29,115]) have enabled researchers to gain deeper insights into the role of m6A in gene regulation, cellular processes, and diseases. Early methods for detecting m6A involved traditional antibody-based approaches. Immunoprecipitation (IP) techniques, such as m6A-RNA immunoprecipitation sequencing (meRIP-Seq or m6A-seq), allowed researchers to enrich m6A-modified RNA fragments using specific antibodies to selectively capture transcripts with bound m6A modifications and then sequence the enriched fragments to identify m6A sites within the transcriptome [116]. To improve the specificity and accuracy of m6A site identification, researchers introduced enhanced cross-linking techniques. m6A cross-linking and immunoprecipitation sequencing (m6A-CLIP-Seq) uses UV cross-linking to lock in the interactions between m6A-modified RNA and binding proteins, offering higher precision in identifying m6A sites [117]. Modified CLIP (miCLIP), which also employs the use of UV cross-linking, induces specific mutations (C to T conversions) during reverse transcription (RT) and enhances the precision of identifying m6A sites [117]. Researchers have developed chemical-based methods that exploit the distinct reactivity of m6A-modified adenosines. These methods involve chemical labeling or selective reactions that target m6A modifications, enabling their enrichment and subsequent sequencing. Liquid chromatography and mass spectrometry (LC-MS) can be used to identify m6A modifications [118] and deamination adjacent to RNA modification targets (DART-Seq) utilizes a fusion protein of the m6A-binding YTH domain and the cytidine deaminase APOBEC1 to guide C-to-U editing at cytidine residues adjacent to m6A sites [119,120]. The emergence of nanopore-based single-molecule sequencing technology has brought about a significant breakthrough in m6A detection. Nanopore sequencing allows for direct detection of modifications as RNA molecules pass through a nanopore. This method eliminates the need for antibodies or chemical labeling, providing a direct and unbiased approach to identifying m6A modifications [121,122,123,124].

To enhance the accuracy and comprehensive understanding of m6A modifications, researchers have begun to integrate multiple techniques. For example, combining miCLIP with RNA sequencing methods and machine learning tools provides a more holistic view of m6A dynamics within the context of RNA transcription and processing [125]. The evolution of m6A detection has led to the integration of m6A data with other omics data, such as transcriptomics, epigenomics, and proteomics. Integrating multiple layers of information offers a more complete understanding of how m6A modifications impact gene expression and cellular processes. Accurate interpretation of m6A-seq data requires sophisticated computational algorithms. Bioinformatics tools have evolved to better predict and annotate m6A sites within RNA sequences, allowing researchers to extract more meaningful insights from their experimental data [124,126,127]. Overall, the evolution of m6A detection methods in recent years has been marked by a transition from traditional antibody-based techniques to more advanced and precise approaches. These advancements have not only improved our ability to identify m6A modifications but have also deepened our understanding of their functional implications in various biological contexts.

## 4. m6A in NAFLD

Relatively little is known about m6A modification in NAFLD. However, emerging studies address the mechanisms of m6A regulation in hepatocellular carcinoma and other liver pathologies (Table 2). Most studies involving m6A and liver disease occurred within the last 4–5 years (Figure 2). Until recently, lack of technologies available to identify and measure transcript-specific m6A presented challenges to understanding the role of m6A modifications in human disease.

The first mention of m6A gene regulation in NAFLD was in 2015. In that study, hepatic SAM concentration was decreased with betaine supplementation in HFD-fed mice, resulting in decreased global m6A methylation and increased FTO expression [156]. SAM plays a role in a number of hepatic cellular pathways (reviewed in [157]). Patients with chronic liver disease and HCC have reduced SAM, due to reduce MAT1A expression in one carbon metabolism, although SAM treatment was insufficient to treat SAM deficiency since the liver upregulated compensatory mechanisms to reduce accumulation and dispose of excess SAM (reviewed in [158,159]). The potential of SAM and betaine as therapeutic interventions in MASLD remains to be fully investigated and the potenial impact of these therapies on the m6A epitranscriptome requires investigation.

The next reference to m6A in the context of NAFLD did not appear until 2020. Knockdown of FTO in a mouse hepatocyte cell line decreased Sterol Regulatory Element Binding Transcription Factor (Srebf) and Scd mRNA and protein and luciferase assays demonstrated that mutation of m6A sites in the 3′UTR of Srebf1 and Scd reduced m6A levels and increased mRNA stability in these lipogenic genes [160]. In a zebrafish model of lipid metabolism, exposure to endocrine-disrupting chemicals (EDCs) resulted in increased hepatic lipid accumulation, a decrease in global m6A, and increased FTO expression [161]. Differences in m6A modification in hepatic genes in HFD-fed mice compared to mice on a control diet were demonstrated using m6A-specific RNA immunoprecipitation sequencing (m6A-RIP-seq, also called MeRIP-seq) [162]. mRNA from the livers of mice fed a HFD had 554 differentially methylated peaks, many of those involved in biological pathways associated with lipid metabolism [162].

In a leptin-receptor deficient obese mouse model there was a global increase in m6A methylation and an increase in the expression of lipogenic genes that correlated to liver steatosis [163]. The obese mouse livers also expressed lower levels of the m6A reader Ythdc2 compared to non-obese mouse livers, suggesting a role for Ythdc2 in decreasing lipid accumulation [163]. Mettl3 transcript abundance was increased in the macrophages from obese mouse livers and myeloid-specific Mettl3-knockout resulted in inhibited HFD-induced obesity in mice with lower hepatic lipid accumulation, inflammation, and fibrosis, demonstrating m6A involvement in immune mechanisms that contribute to NAFLD and progression to NASH [164]. In a study comparing two NAFLD models (leptin-deficient db/db mice and high fructose fed (HFrD)-mice), both groups had increased lipid metabolism compared to normal diet control, however, the HFrD mice had an increased inflammatory response and increased m6A methylation of FASN mRNA [165] demonstrating diet-specific differences in m6A regulation of inflammation. In a HFD-fed mouse model, upregulation of Mettl3 and Ythdf1 and a global increase of m6A were noted in hepatocytes [166]. Knockdown of Mettl3 or Ythdf1 decreased Rubicon, an autophagosome-lysosome fusion suppressing protein, reducing clearance of autophagosomes, indicating a role for m6A and m6A-RBPs in regulating autophagy in NAFLD [166]. In a HFD-induced NAFLD mouse model, infection with an adenovirus expressing FTO (ad-FTO) increased mRNA and protein expression of SREBF1 and ChREBP [167]. SREBF1 and ChREBP stabilized these transcripts and promoted lipogenesis [167]. In a mouse model of CCl_4_-induced liver fibrosis, global m6A levels were increased in fibrotic livers compared to normal mouse livers and this m6A increase was inversely correlated to a significant decline in ALKBH5 expression [168]. Drp1 encodes a protein that mediates mitochondrial fission in HSC cells, promoting proliferation and migration, and YTHDF1 bound to Drp1 (gene Dnm1l, dynamin 1-lik2) mRNA at an m6A motif and promoted Drp1 translation [168]. Furthermore, forced expression of ALKBH5 in HSC-T6 immortalized rat hepatic stellate cells correlated with upregulation of Drp1 [168]. Hepatic stellate cell (HSC) activation promotes liver fibrosis through the stimulation of extracellular matrix production [169]. In a study using HSC-specific Mettl3 knockout mice, the expression of pro-fibrotic genes, including Acta2, Col1a1, and Timp1 were decreased with Mettl3 inhibition, suggesting a HSC-specific role for m6A in liver fibrosis [170].

A genome-wide expression study compared the expression of m6A writers, readers, and erasers in liver biopsies from NAFLD patients and healthy controls [171]. NAFLD liver samples had increased METTL3, METTL15, FTO, and EIF3H (an m6A writer) expression, while WTAP, RBM15, YTHDC1, YTHDC2, IGF2BP2, HNRNPC, and HNRNPA2B1 expression was decreased [171]. EIF3H was correlated to steatosis and VIRMA was correlated to degree of lobular inflammation [171]. Additionally, the authors performed co-expression analysis of the m6A reader RNA binding proteins (RBPs) and found strong interaction relationships between RBM15, HNRNPC, YTHDC2, and HNRNPA2B1 [171]. Gene set enrichment analysis (GSEA) of the co-expressed m6A regulators showed 33 KEGG pathways that were significantly enriched including, steroid hormone biosynthesis, cytochrome P450, nucleotide metabolism, and MAPK signaling [171]. The authors concluded that this analysis highlighted the effects of dysregulated m6A on steatosis and fibrosis in NAFLD.

Other exposures have been associated with altered liver m6A. For example, global hepatic m6A was significantly increased in Western diet-fed mice treated with vinyl chloride (VC) [172]. Importantly, the VC-exposed mice developed more severe steatotic liver disease and HCC at a high frequency. In addition to m6A, several other epitranscriptomic marks were significantly changed by VC exposures. We recently published on modified nucleotides and bases circulating in the blood or urine of human subjects with alcohol-associated liver disease (ALD). While numerous modified bases were significantly altered, m6A was unchanged [173]. The abundance of modified nucleobase and ribonucleoside, 7,9-dimethylguanine in urine and 2-methylthio-N6-threonylcarbamoyladenosine (ms2t6A) in serum were strongly associated with the severity of the ALD. However, m6A has been used to subtype human alcohol-related HCCs [174]. While more data are required, there appears to be an emerging relationship between environmental exposures, steatotic liver diseases, and hepatocellular carcinomas.

Taken together, these studies suggest that m6A modification affects the stability and translation of key genes involved in hepatic lipid metabolism, inflammation, and fibrosis by linking the dysregulation of m6A modification and m6A modifiers to the accumulation of fat in the liver, inflammation, and oxidative stress, all of which are key features of NAFLD. While some progress has been made in understanding the role of m6A in NAFLD, there are several areas that remain to be studied. Areas for future study include the identification of specific m6A-modified transcripts and pathways that are involved in the pathogenesis of NAFLD, validation of m6A modifications as a diagnostic or prognostic biomarkers for NAFLD, the development of therapeutic strategies targeting dysregulated m6A modifications in NAFLD, and the potential crosstalk between m6A modification and other post-transcriptional modifications, including alternative splicing and miRNA regulation.

## 5. m6A and Liver Physiology

### 5.1. m6A and Lipid Metabolism

Disruption of the balance between fatty acid uptake, de novo lipogenesis, and lipid oxidation leads to excess fat accumulation in the liver which is a hallmark of NAFLD [175,176,177]. Recent studies have recognized the role of m6A writers, erasers, and readers in the regulation of genes responsible for fatty acid oxidation [178], lipogenesis [179], and lipid transport [180]. In hepatic lipid metabolism disorders, genes with m6A hypermethylation in HFD-induced fatty livers are enriched in processes associated with lipid metabolism, whereas hypo-methylated m6A sites are linked to translation-related processes [162]. In a mouse model of HFD and toxicant exposure, genes with PCB and HFD-induced m6A changes were enriched in pathways related to lipid and lipoprotein metabolism [181]. Furthermore, m6A sequencing conducted on adult pig livers demonstrated that highly m6A-containing transcripts were enriched in pathways related to the positive regulation of metabolic processes and fatty acid transport [182]. In the livers of mice with specific knockout of the Mettl3 gene in hepatocytes, there was a significant reduction in the expression of genes which regulates fatty acid synthesis and oxidation, including Ehhadh, Fasn, Foxo1, Ppargc1a, and Sirt1, and also decreased circadian clock Bmal1 and Clock transcript and protein expression [183]. The hepatocyte-specific deletion of Mettl3 resulted in improvements in insulin sensitivity and glucose homeostasis, indicating that these effects were dependent on m6A modification [183].

Additionally, the circadian clock influences hepatic lipid metabolism by affecting the decay of peroxisome proliferator-activated receptor-α (PPARα) mRNA through YTHDF2 mediation [184], establishing a connection between the circadian clock and metabolic diseases. Another study showed that carbon tetrachloride (CCl_4_)-induced liver fibrosis in mice resulted in increased total liver m6A with specific increased m6A in NR1D1 [49]. Three changes in RWE were detected at the mRNA transcript level in CCl_4_-exposed livers: increased ZC3H13 and YTHDC1 and decreased FTO. NR1D1 encodes the ‘orphan nuclear receptor’ REV-ERBα, a transcriptional repressor that regulates diurnal de novo lipogenesis in the liver [185]. RNA immunoprecipitation showed that the NR1D1 transcript associated with YTHDC1 in the CCl_4_-induced fibrotic liver and in hepatic stellate cells (HSC) [49]. The investigators identified one m6A position in the coding sequence of the NR1D1 transcript. They cloned a 22 nt fragment with the A or a G in the DRACH motif into a luciferase reporter and showed that siYTHDC1 co-transfection of HSC-LX2 cells resulted in increased luciferase activity only when A was present. These data were suggested to indicate that YTHDC1 binding m6A in the NR1D1 transcript resulted in transcript degradation and reduced NR1D1 protein in the CCL4-exposed livers. The decrease in NR1D1 was associated with de-phosphorylation of the mitochondrial fission protein DRP1, increased reactive oxygen species (ROS), reduced mitochondrial fission, mitochondrial DNA release, and activation of cyclic GMP-AMP synthesis (cGAS) inflammatory signaling, HSC activation, and hepatic fibrosis [49]. However, how NR1D1 regulated DRP1 phosphorylation was not elucidated.

FTO is recognized for its impact on adipogenesis and metabolism. FTO expression is significantly higher in the livers of rats with NAFLD and patients with NASH [186,187]. In T2DM patients, glucose increases FTO expression, and decreases global m6A content [188]. In mouse models, FTO depletion correlates to a significant decrease in body weight and adipose tissue, while mice with increased levels of FTO exhibit heightened food intake [189]. When FTO is overexpressed in HepG2 cells, the expression of genes involved in lipid metabolism such as FASN, SCD1, and MGAT1 are increased [190]. Conversely, FTO reduces the expression of genes associated with lipid transport such as MTTP, APOB, and LIPC, which results in lipid accumulation [190]. However, these effects are not observed when the FTO R316A mutant, lacking demethylase activity, is present [190]. Additionally, the overexpression of FTO can elevate the levels key lipogenic regulators, SREBP1c and CIDEC in hepatocytes [191]. Importantly, using short hairpin RNA (shRNA) to knock down FTO diminished dexamethasone-induced fatty liver in mice [160]. In summary, these findings indicate that under lipotoxic conditions, FTO may have a detrimental role in hepatocytes by modifying m6A patterns. These studies suggest that the dynamic process of m6A methylation may play a crucial role in liver metabolism (Figure 3).

### 5.2. m6A and Glucose Metabolism

Aberrant glucose metabolism has emerged as a key player in the development and progression of NAFLD. Glucose metabolism is a tightly regulated and is disrupted by insulin resistance, a hallmark feature of NAFLD [193,194]. Insulin resistance increases production of glucose in the liver (gluconeogenesis), further exacerbating hyperglycemia [195]. Further, dysregulated glucose metabolism in NAFLD is closely associated with lipogenesis [193]. Elevated levels of glucose promote the hepatic synthesis of fatty acids through de novo lipogenesis, leading to the accumulation of triglycerides within hepatocytes [193]. The excess triglycerides further contribute to hepatocellular lipid accumulation, leading to the characteristic hepatic steatosis observed in NAFLD [196]. A study using mouse hepatocytes demonstrated that depletion of *Mettl3* reduced glycogen storage by decreasing m6A levels on the glycogen synthase 2 (*Gys2*) transcript which is stabilized by IGF2BP2 [153]. In another study, mice fed a HFD had increased *Mettl3* transcript expression and higher m6A levels which correlated to altered expression of glucose metabolism genes including *Lpin1*, a liver metabolism regulator [197]. Conversely, the expression of m6A erasers, FTO and ALKBH5, was decreased in obese and T2DM patients compared to healthy controls [198]. Additional m6A readers have been implicated in the regulation of glucose metabolism. For example, YTHDF3 was reported to promote glycolysis by increasing phosphofructokinase (*PFKL*) expression in HCC cells [150]. Additionally, YTHDC1 has been implicated in the regulation of glucose metabolism and insulin resistance by interacting with serine/arginine splicing factor (*Srsf3*) and cleavage and polyadenylation specific factor 6 (*Cpsf6*) to regulate mRNA splicing of in mouse β-cells [199]. Prior to the discovery of their role as m6A readers, IGF2BPs were identified by genome-wide association studies as T2DM associated genes [200,201,202]. As a result of conserved IGF2BP binding sites enriched in “GGAC”, there is considerable overlap with the m6A motif, strengthening preferential recognition of m6A-modified mRNAs [79,203,204]. These findings suggest that m6A RWE modulate glucose utilization by affecting key metabolic genes or signaling pathways involved in glucose metabolism.

### 5.3. m6A and Hepatic Stellate Cell Activation

Hepatic stellate cells (HSC) are undifferentiated cells that play a role in liver regeneration by differentiating into endothelial cells in response to liver damage [205]. HSC activation is a driver of liver fibrosis [206], a critical step in developing cirrhosis [207,208]. As activated HSCs develop into myofibroblasts, they deposit excess extracellular matrix which results in the production of scar tissue contributing to liver fibrosis [206]. Recent studies have identified potential mRNA targets of m^6^A and related pathways using in vivo and in vitro models of HSC activation [209,210,211]. In a CCl_4_-induced mouse model of liver fibrosis, 3315 genes had altered m6A levels and genes with differential m6A were enriched in pathways associated with liver fibrosis and HSC activation, including; endoplasmic reticulum (ER) stress, PPAR signaling, and TGF-β signaling [47]. In another study, ALKBH5 was found to be decreased in fibrotic mouse livers which resulted in increased HSC activation. Conversely, overexpression of ALKBH5 inhibited HSC activation and suppressed liver fibrosis [209]. Transforming growth factor-β1 (TGF-β1) is a fibrogenic cytokine commonly used to activate HSC in vitro [212]. Liver fibrosis is increased in Prdx3 (peroxiredoxin 3) knockdown mice; however, mutation of three m6A sites in the *Prdx3* transcript resulted in a loss of YTHDF3-mediated down-regulation of PRDX3 protein, thus increasing PRDX3 [91]. Although no studies have shown a direct interaction between m6A and TGF-β1 expression, loss of m6A methylation mediated by FTO and RALY RNA binding protein-like (RALYL), increased the stability of TGF-β2 in HCCs promoting cancer stemness and cell proliferation [213].

### 5.4. m6A and Hepatic Immune Response Signaling

The accumulation of lipids in NAFLD promotes lipotoxicity in the liver, resulting in upregulated immune response signaling and the recruitment of immune cells, inflammasomes, and macrophages [214]. Kupffer cells (KCs) are hepatic macrophages, located in liver sinusoids, that contribute to activation of HSCs and the progression from NAFL to NASH by secreting TGF-β1 [215]. KCs are activated in response to hepatic injury, inflammation, and damage [216] and produce tumor necrosis factor-α (TNF-α), interlukin-1β (IL-1β), and reactive oxygen species (ROS) [217,218,219]. In lipopolysaccharide (LPS)-induced activation of KCs, overexpression of METTL3 and global m6A hypermethylation increased TGF-β1 secretion [220]. In response to radiation, hTERT (telomerase reverse transcriptase)-HSC cells had increased expression of HMGB1, protein that induces interferon expression. In irradiated hTERT-HSC cells, ALKBH5 demethylated HMGB1 mRNA, and inhibited YTHDF2-mediated transcript degradation, demonstrating that HMGB1 transcript stability was regulated by m6A methylation [221]. A study observing m6A modification on inflammatory signaling pathways in piglets, found that LPS initiated an inflammatory response through the NOD1/NFкB signaling pathway and caused dramatic changes in m6A modifications of transcripts in NFкB, Toll-like receptor, and HIF-1 signaling, suggesting that m6A regulates hepatic immune response signaling [222].

## 6. Other mRNA Modifications in Liver

Three common mRNA transcript modifications [36] are less well-characterized than m6A and are included here to provide the reader references for further reading.

### 6.1. Pseudouridine (Ψ)

Pseudouridine is the most abundant modified nucleotide in RNA and is enriched in ncRNAs, tRNAs, and has also been identified in mRNA, with 74% in introns, in human HepG2 hepatocellular carcinoma cells [223]. Ψ was round enriched near alternative splice regions and splice sites where RBPs, i.e., U2AF2, U2AF1, SF3A3, PRPF8, HNRNPC, PTBP1, and were shown to bind [223]. Knockout of the pseudouridine synthases PUS1 and PUS2 altered splicing of distinct pre-mRNA targets in HepG2 cells [223]. The role for Ψ modification of mRNA transcripts in NAFLD is unexplored.

### 6.2. 1-Methyladenosine (m1A)

The m1A modification is found in rRNA, tRNA, and mRNA is catalyzed by the writers TRMT61A and TRMT6, read by HRSP12, and erased by FTO, ALKBH5, and ALKBH3. m1A methylation is increased in human HCC tumors and in liver cancer stem cells (CSC) [224]. Increased m1A in tRNAs was reported to increase PPARγ translation in CSCs to increase cholesterol synthesis, activate Hedgehog (Hh) signaling, and drive self-renewal of liver cancer stem cells (CSC) in HCC [224]. No studies on m1A in NAFLD were found.

### 6.3. 5-Methylcytidine (m5C)

m5C is found in mRNA, tRNA, and rRNA. Cholangiocarcinoma (CCA) is the second most common primary liver cancer and has poor prognosis [225]. In cholangiocarcinoma (CCA) cell lines, the lncRNA NF-kappa B interacting lncRNA (NKILA), which is overexpressed and associated with reduced overall survival of CCA patients, was modified by m5C which stabilized NKILA [226]. Mechanistically, NKILA was identified as a competing endogenous RNA (ceRNA) for miR-582-3p; thus ‘sponging miR-582-3p, and relieving its repression of YAP1 protein expression, a driver of CCA [227].

## 7. Conclusions

Data reported in the literature support the conclusion that changes in m6A in mRNA alter transcript stability of genes involved in liver pathologies and relate to genes and pathways in hepatic regulation of glucose and fat metabolism. Further, expression of the m6A METTL3 writing complex, erasers (demethylases: FTO and ALKBH5), and m6A readers (RBPs) are altered in human NAFLD and in experimental models of liver disease and contribute to NAFLD, HCC, and CCA. Further studies are needed to investigate the specific mechanisms by which mRNA transcript-specific m6A interact with RBPs and alter translation and RNA degradation in the liver and the mechanisms regulating the levels of m6A RWE in response to diet and environmental-induced liver disease.

## Figures and Tables

**Figure 1 genes-14-01653-f001:**
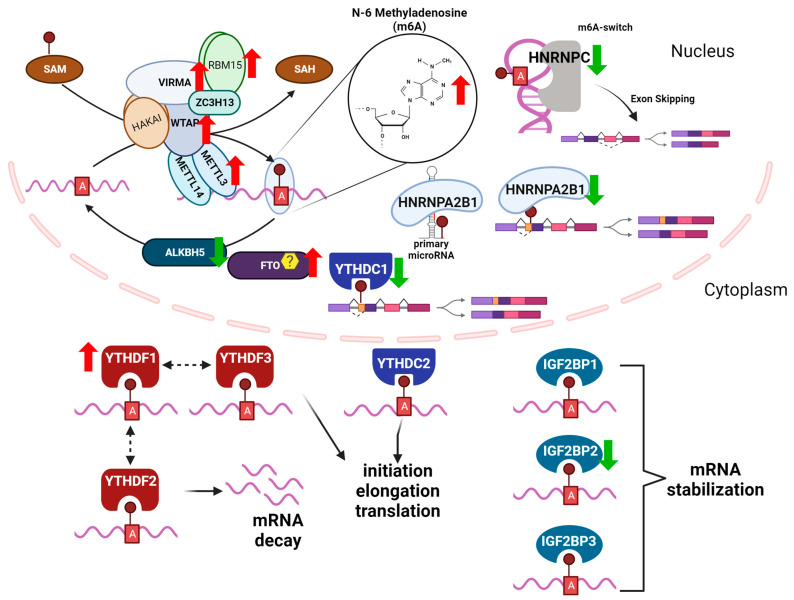
m6A readers, writers, and erasers (RWE) in NAFLD, based in part on [74] RWE of m6A in mRNA regulate transcript fate. Shown is the METTL3 m6A methyltransferase (writer) complex which includes: METTL3, METT14, WTAP, VIRMA, ZC2H13,RBM15,and HAKAI and the predominant subcellular location and roles of reader proteins, including nuclear RBPS: HNRNPA2B1, HNRNPC, and YTHDC1 and cytoplasmic RBPs: YTHDF1, YTHDF2, YTHDF3, YTHDC2, IGF2BP1, IGF2BP2, and IGF2BP3. ALKBH5 is a specific m6A demethylase, whereas there is controversy as to the specificity of FTO for m6A demethylation as described in the text. The up (red) and down (green) arrows indicate the expression levels generally identified in hepatic disease conditions (see further details in Table 1). Created with BioRender.com.

**Figure 2 genes-14-01653-f002:**
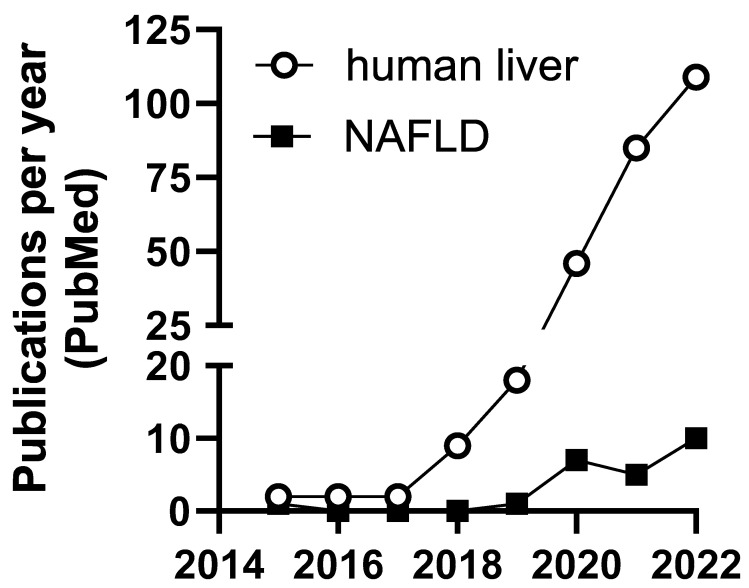
Number of publications per year including m6A and liver or NAFLD in PubMed.

**Figure 3 genes-14-01653-f003:**
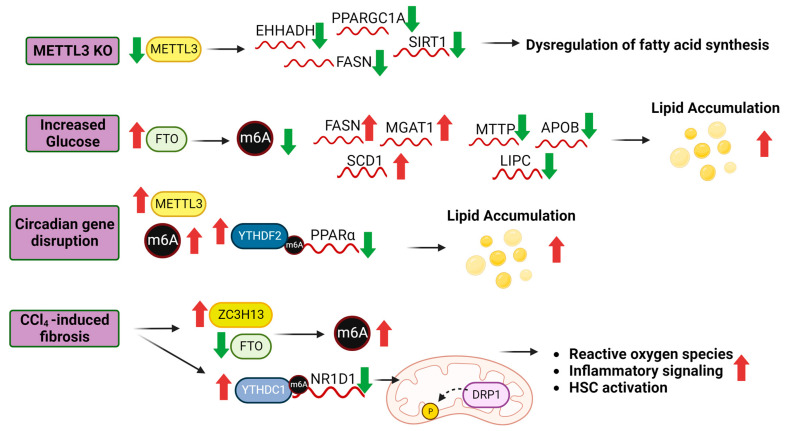
Role of m6A modifications in lipid metabolism. The liver expression of m6A RWE is perturbed by physiological and disease processes, resulting in changes in m6A levels and subsequently, transcript processing and protein translation regulated in part by m6A readers. Experimental hepatocyte-specific knockout of Mettl3 in mice increased fibrosis and steatosis and downregulated the indicated transcripts [192]. In T2DM patients, glucose increases FTO expression [188]. Overexpression of FTO in HepG2 cells altered the expression of the indicated transcripts which resulted in increased lipid accumulation [190]. Circadian gene disruption in mice increased Mettl3 and Ythdf2 expression which mediated Ppara mRNA decay [184]. CCl_4_-induced liver fibrosis in HSCs increased ZC2H13 and decreased FTO increasing m6A levels. YTHDC1 bound to m6A on the NR1D1 transcript, decreasing NR1D1 expression which resulted in disrupted fatty acid regulation, lipid accumulation and liver inflammation [49]. Abbreviations: Enoyl-CoA Hydratase And 3-Hydroxyacyl CoA Dehydrogenase (EHHADH), PPARG Coactivator 1 α (PPARGC1A), Sirtuin 1 (SIRT1), Fatty Acid Synthase (FASN), α-1,3-Mannosyl-Glycoprotein 2-β-N-Acetylglucosaminyltransferase (MGAT1), Stearoyl-CoA Desaturase (SCD1), Microsomal Triglyceride Transfer Protein (MTTP), Apolipoprotein B (APOB), Lipase C, Hepatic Type (LIPC), Peroxisome Proliferator Activated Receptor α (PPARα), Nuclear Receptor Subfamily 1 Group D Member 1 (NR1D1). Further details from these studies are provided in the text.

**Table 1 genes-14-01653-t001:** Key functions of m6A in mRNA metabolism.

Function	Description	Examples
mRNA stability	m6A can influence RNA stability by either increasing or decreasing the half-life of the transcript.	m6A increased the stability of the circadian clock gene transcript *NR1D1*, activating hepatic stellate cells (HSC) in a liver fibrosis cell model [49]. m6A decreased the stability of *IFNB* in human foreskin fibroblasts and m6A depletion enhanced *IFNA* and *IFNB* mRNA in response to viral infection [50].
mRNA processing	m6A affects mRNA polyadenylation and other processing events. The presence of m6A near alternative splicing sites can enhance or inhibit the recognition of the spliceosome, leading to changes in the splicing pattern of the pre-mRNA.	In a cell model of pancreatic ductal adenocarcinoma, HNRNPC interacted with m6A sites in TAF8 to increase exon skipping and increased the transcript abundance of the pro-metastatic isoform TAF8S [51]. HCV infection induces the loss of an m6A peak on CIRBP, a gene that encodes for a stress-induced binding protein, in HepG2 cells. The decreased m6A levels resulted differential CIRBP isoform, suggesting a role for HCV-induced m6A changes on alternative splicing [52].
mRNA localization	m6A can affect the localization of mRNA transcripts within cells	In mouse hippocampal neurons, Mettl3 knockout reduced m6A in the 3′UTR of Camk2a and Map2, inhibiting mRNA localization and reducing transcript abundance in neurites [53]. Knockdown of the m6A reader YTHDC1 in HeLa cells reduced cytoplasmic abundance of mRNA [54].
mRNA translation	m6A affects the translation efficiency of mRNA transcripts.	Knockdown of the m6A eraser FTO increased m6A in axonal GAP-43 mRNA and decreased GAP-43 translation in cultured dorsal root ganglia neurons [55]. In HEK-293 cells, METTL3 knockdown decreased m6A in coding regions of transcripts, lead to ribosomal pausing. and decreased translation efficiency [56]. In human gastric cancer cells, translation efficiency of USP14 was increased by YTHDF1 and dependent on m6A methylation in the 3′UTR of the USP14 transcript [57].

**Table 2 genes-14-01653-t002:** Evidence of m6A function in liver pathologies. The role of m6A modification and specific m6A regulators (readers, writers, and erasers) in liver disease.

m6A Function in Liver Disease	M6A Modifiers
m6A levels in Circ-CCT3, a circular RNA upregulated in human HCC tissue that promoted growth and migration of HCC cells by sponging miR-378a-3p, were decreased by METTL3 knockdown and increased by ALKBH5 knockdown. Additionally, knockdown of METTL3 increased Circ-CCT3 expression in HCC cells [128].	ALKBH5 (eraser), METTL3 (writer)
m6A transcriptome-wide profiling in livers of C57BL/6J mice identified 256 differentially methylated peaks in mRNA transcripts in pathways associated with hepatic ischemia reperfusion [129].	
PPARGC1A plays an anti-oncogenic role in human and rodent models [130]. Knockdown of METTL3 increased PPARGC1A mRNA by decreasing four m6A sites near the stop codon of the PPARGC1A 3′UTR in Huh7 cells and YTHDF2 bound directly to PPARGC1A mRNA and promoted degradation of the transcript [131].	METTL3 (writer), YTHDF2 (reader)
FZD10 induced liver cancer stem cell (CSC) expansion. Knockdown of METTL3 or YTHDF2 reduced FZD10 mRNA in HCC cells [132].	METTL3 (writer), YTHDF2 (reader)
Mettl3 and m6A levels were upregulated in lenvatinib-resistant Huh7 cells compared to parental cells and Mettl3 promoted EGFR translation and lenvatinib resistance [133].	METTL3 (writer)
LRPPRC upregulated PD-L1 by increasing m6A modifications in HepG2 and Hep3B cells. Knockdown of LRPPRC increased apoptosis and reduced migratory ability of HCC cells [134].	LRPPRC (reader)
lncRNA miR4458HG was upregulated in human HCC tissue samples and activated the glycolysis pathway in HCC cells. miR4458HG interacted with IGF2BP2 to stabilize SLC2A1 and HK mRNA in BEL-7404 cells [135].	IGF2BP2 (reader)
CircCCAR1 was increased in human HCC tissues and promoted tumor growth in a xenograft mouse model. m6A levels in CircCCAR1 were increased by WTAP and the interaction between WTAP and IGF2BP3 stabilized CircCCAR1 [136].	WTAP (writer), IGF2BP3 (reader)
NRD1 deficiency promoted liver fibrosis in a CCl_4_-induced mouse liver fibrosis model. Additionally, ZCH3H13 was decreased and FTO was increased. Knockdown of ZCH3H13 and overexpression of FTO in HSC-LX2 cells decreased m6A methylation of NR1D1 mRNA, preventing YTHDC1 binding and stabilization of the NR1D1 transcript [49].	ZCH3H13 (writer), FTO (eraser)
KIAA1429 was upregulated in sorafenib-resistant HCC cells and KIAA1429 depletion inhibited cell invasion and migratory ability [137]. KIAA1429 is elevated in HCC tissues and is associated with reduced overall survival [138]. KIAA1429 promotes m6A methylation of the 3′UTR of GATA3 pre-mRNA in the nucleus of SK-Hep1 and HCCLM3 cells resulting in transcript degradation and reduced GATA3 protein [138].	KIAA1429 (writer)
RBM15 expression was upregulated and global m6A was increased in mouse fetal liver tissue of Gestational diabetes mellitus offspring and overexpression of RBM15 increased insulin resistance in primary mouse hepatocytes [139].	RBM15 (writer)
Post-translational O-GlcNAcylation of YTHDF2 at Ser263 contributed to HBV-related hepatocarcinogenesis by promoting stabilization of MCM2 and MCM5 mRNA in HepG2-HBV1.3 cells [140].	YTHDF2 (reader)
MeRIP-seq of LX2 human HSCs with ALKBH5 knockdown revealed fewer m6A peaks and fewer m6A peak-containing genes that the LX2 control. Silencing of ALKBH5 also downregulated CCL5, however silencing YTHDF2 restored CCL5 expression and promoted monocyte recruitment and polarization of irradiated LX2 cells [141].	ALKBH5 (eraser), YTHDF2 (reader)
YTHDF2 regulated myeloid cell homeostasis in immune hepatitis through targeted degradation of Rxra in mouse myeloid-derived suppressor cells (MDSCs). YTHDF2 depletion increased MDSC expansion and decreased apoptosis [142].	YTHDF2 (reader)
m6A modifications in HDAC1 mRNA was increased in the livers of rats with diet-induced metabolic syndrome. Additionally, METTL3 was upregulated and RIP-assays revealed that IGF2BP2 bound to HDAC1 mRNA and increased HDAC1 expression [143].	METTL3 (writer), IGF2BP3 (reader)
Upregulation of VIRMA in intrahepatic cholangiocarcinoma (ICC) cells increased cell proliferation and invasion. CCL3, a cytokine secreted by hepatocytes, interacted with VIRMA to alter m6A modifications in ICC cells, specifically upregulating SIRT1, a downstream target of VIRMA-mediated m6A modification [144].	VIRMA (writer)
A genome-wide analysis of tumors from HCC patients demonstrated an association between specific oncogenic lncRNAs and m6A modification. The correlations between lncRNA and m6A were identified as prognostic markers that can be used in HCC risk-assessment [145].	
m6A-RIP-seq on human liver samples revealed higher hepatic m6A and increased expression of m6A writers, including METTL3, METTL14, and WTAP in biliary astresia (BA) patients compared to normal human liver controls. Additionally, m6A levels were increased in BA patients with advanced stage fibrosis, compared to early stage BA patients [146].	METTL3(writer), METTL14(writer), WTAP (writer)
SLP2, a prognostic marker for HCC, was decreased in HCC cells with the inhibition of METTL3 and METTL3 is positively correlated with SLP2 in HCC patients [147].	METTL3 (writer)
PARK7 mRNA encodes the Ras-dependent oncoprotein DJ-1 which is significantly upregulated in HCC patients. Overexpression of WTAP in HCC cells increased m6A modifications in PARK7 and IGF2BP1 bound PARK7 at an m6A site, stabilizing PARK7 mRNA [148].	WTAP (writer), IGF2BP1 (reader)
The oncogenic circRNA, circFUT8 was upregulated in HCC and METTL14 promoted the m6A modification of circFUT8 in HCC cells. Additionally, M1 macrophage-derived exosomal miR-628-5p reduced m6A modification of circFUT8 be reducing METTL14 expression in HCC cells [149].	METT14 (writer)
YTHDF3 facilitated HCC progression by promoting glycolysis in HCC cells and preventing the degradation of phosphofructokinase (PFKL) mRNA, via binding to the m6A modification in the PFKL transcript [150]	YTHDF3 (reader)
After partial hepatectomy in C57BL/6 mice, global liver m6A levels were increased and were correlated to an increase in METTL14 and hepatocyte growth factor (HGF) expression. The hepatic regenerative ability of hepatocyte-specific Mettl14 knockout mice was significantly reduced compared to wild-type [151].	METTL14 (writer)
m6A regulators, i.e., YTHDC1, RBM15, and METTL3 were associated with HCC stage in human HCC tissue. While they were found to be upregulated in early stages of HCC, the expression of these m6A regulators was decreased in stage 4 HCC, suggesting that m6A modifications are altered in an HCC stage-specific manner [152].	YTHDC1(reader), RBM15(reader), METTL3 (writer)
Global inhibition of METT3 expression in livers from Mettl3 knockout (ko) mice showed reduced expression of AQP8, a channel protein associated with glycogen accumulation. STM2457-inhibition of METTL3 activity decreased m6A levels and decreased glycogen storage capacity in mouse hepatocytes [153].	METTL3 (writer)
The expression of METTL16 was significantly higher in livers from chronic hepatitis B patients with severe fibrosis compared to those with only mild fibrosis and METTL16 was positively correlated the expression of genes associated with chronic hepatitis B (CHB), including HLA-DPB1 and HLA-DPA1 [154].	METTL16 (writer)
Knockdown of METT14 in HepG2 cells resulted in the differential m6A modification of 8 lncRNAs associated with HCC in liver patients. ARHGAP5-AS1 had the most m6A changes. RIP-qPCR assays in HCC cells demonstrated that IGF2BP2 had the highest binding affinity with ARHGAP5-AS1 and IGF2BP2 deletion decreased ARHGAP5-AS1 expression [155].	METTL14 (writer), IGF2BP2 (reader)

## Data Availability

Not applicable.

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
