# Peer review of "Changes in m6A in Steatotic Liver Disease"

_genes, 2023, doi:10.3390/genes14081653_

Round 1

Reviewer 1 Report

The manuscript by Petri et al. entitled “Changes in m6A in Steatotic Liver Disease” reviewed writers, readers, and erasers of m6A and current methods of detecting m6A in specific genes associated with fatty liver disease and hepatocellular carcinoma (HCC). Overall, the paper is well-written and interesting, but there are still some suggestions to improve the quality of the manuscript.

1. It would be better for the readers to understand if there are more summary figures in the paper.

2. The detecting methods of m6A should be included in the paper.

3. In figure 1, authors should consider revising and expanding the diagram to explain how m6A-related genes participate in the process of metabolic dysfunction-associated steatotic liver disease.

4. Figure 2 is too simple. In table 1, m6A affect translation is missing.

5. Although the authors mention m6A research in HCC, this part is poorly summarized. I suggest that the authors should delete this part and focus on Steatotic Liver Disease.

Author Response

Thank the Editor for the opportunity to submit our revised review manuscript: “Changes in m6A in Steatotic Liver Disease” for review. 

We have addressed each the comments from the two reviewers. The Reviewers’ comments are in Calibri 11 and our responses are in Arial 11 bold.

Response to Reviewer 1

The manuscript by Petri et al. entitled “Changes in m6A in Steatotic Liver Disease” reviewed writers, readers, and erasers of m6A and current methods of detecting m6A in specific genes associated with fatty liver disease and hepatocellular carcinoma (HCC). Overall, the paper is well-written and interesting, but there are still some suggestions to improve the quality of the manuscript

  1. It would be better for the readers to understand if there are more summary figures in the paper.

A new figure (figure 3) has been added to illustrate the roles of m6A and m6A readers, writers, and erasers in lipid metabolism.

  1. The detecting methods of m6A should be included in the paper.

We thank the reviewers for this suggestion and have added a section on m6A detection methods.

  1. In figure 1, authors should consider revising and expanding the diagram to explain how m6A-related genes participate in the process of metabolic dysfunction-associated steatotic liver disease.

We have included red and green arrows indicating the direction of the expression of RWE in liver disease in Figure 1. 

  1. Figure 2 is too simple. In table 1, m6A affect translation is missing.

We respectfully note that given the low number of publications on m6A in liver and in NAFLD, Figure 2 is simple and will challenge readers to expand their research to contribute the understanding of m6A in MASLD and other liver diseases.  With respect to the second line, we thank the Reviewer for recognizing this omission, we have edited the Table 1 to include m6A role in translation.

  1. Although the authors mention m6A research in HCC, this part is poorly summarized. I suggest that the authors should delete this part and focus on Steatotic Liver Disease.

We agree that while mention of m6A in HCC is important in this review, it is not the focus; thus, we removed HCC from line 23 in the introduction stating that this was part of the goal of this review.

Reviewer 2 Report

The manuscript “Changes in m6A in Steatotic Liver Disease” represents an interesting and valuable review article describing the readers, writers, and erasers (RWE) of m6A and current methods of detecting m6A in specific genes associated with fatty liver disease and hepatocellular carcinoma (HCC).

However, some obstacles must be removed before the manuscript is ready for publication.

These, among others, include:

Line 72… The authors have written: “Demethylases FTO and ALKBH5”. This sentence seems unfinished and should be either deleted or reformulated-finished.

The synonyms like FTO should be followed by their full name in parenthesis when mentioned for the first time in the manuscript text.

The manuscript should be accompanied by a corresponding list of abbreviations and their description. Also, it will be helpful to have a description of the synonyms in the figure legends when appropriate.

The readers, writers, and erasers in Figure 1 should be designated by different colors/shades of it for more effortless follow-up, or writers, readers, and erasers should be grouped and enumerated in the figure legend.

The authors should always use the exact synonyms for particular genes or proteins. For example, it should be either FTO or Fto, not both of them. Also, the usual way of presenting gene (italics) and protein synonyms in the text should be followed and always written in capital letters or otherwise, but not using both ways for the same gen or protein.

Line 160-161… The sentence “There is evidence that ALKBH5 is the specific m6A-demethylase whereas FTO is the m6Am demethylase” is confusing and should be reformulated. The term specific should be explained.

The authors have used the exact enumeration for tables (entitled Key functions of m6A in mRNA metabolism and Evidence of m6A function in liver pathologies) designated as Table 1, which must be corrected.

The table entitled Evidence of m6A function in liver pathologies should be revised by enlisting writers, readers, and erasers separately. Also, some of the text parts in that table should be explained in the manuscript text rather than in the table (the parts without appropriate writers, riders, and erasers).

 The authors should more carefully subdivide the manuscript text into appropriate subsections/paragraphs. For example, the text encompassed by lines 297 to 313 should be separated from the previous text.

A major revision of the manuscript is recommended.

Author Response

Response to Reviewer 2

Comments and Suggestions for Authors

The manuscript “Changes in m6A in Steatotic Liver Disease” represents an interesting and valuable review article describing the readers, writers, and erasers (RWE) of m6A and current methods of detecting m6A in specific genes associated with fatty liver disease and hepatocellular carcinoma (HCC). However, some obstacles must be removed before the manuscript is ready for publication. These, among others, include:

Line 72… The authors have written: “Demethylases FTO and ALKBH5”. This sentence seems unfinished and should be either deleted or reformulated-finished. The synonyms like FTO should be followed by their full name in parenthesis when mentioned for the first time in the manuscript text.

The incomplete line “Demethylases FTO and ALKBH5” has been removed. FTO has been defined at first mention.

The manuscript should be accompanied by a corresponding list of abbreviations and their description. Also, it will be helpful to have a description of the synonyms in the figure legends when appropriate.

A list of commonly used abbreviations has been added to the end of the manuscript

The readers, writers, and erasers in Figure 1 should be designated by different colors/shades of it for more effortless follow-up, or writers, readers, and erasers should be grouped and enumerated in the figure legend.

The Figure 1 legend has been edited to include the designations for each RBP as part of the m6A writer, complex, readers proteins, or erasers (demethylases).

The authors should always use the exact synonyms for particular genes or proteins. For example, it should be either FTO or Fto, not both of them. Also, the usual way of presenting gene (italics) and protein synonyms in the text should be followed and always written in capital letters or otherwise, but not using both ways for the same gen or protein.

We agree with the reviewer.  Instances where genes are identified have been italicized and proteins are written in all capital letters. Only when studies referenced are specific to mouse genomes are genes written as uppercase first letter and the gene abbreviation is italicized.

Line 160-161… The sentence “There is evidence that ALKBH5 is the specific m6A-demethylase whereas FTO is the m6Am demethylase” is confusing and should be reformulated. The term specific should be explained.

This line has been rewritten to clarify that ALKBH5 is the demethylase responsible for removing m6A, “There is evidence that ALKBH5 is the demethylase specifically responsible for removing the methyl group from m6A -modified mRNA, whereas FTO is the m6Am demethylase”.

The authors have used the exact enumeration for tables (entitled Key functions of m6A in mRNA metabolism and Evidence of m6A function in liver pathologies) designated as Table 1, which must be corrected.

Thank you for pointing out this oversight. Table numbers have been corrected to have unique identifiers.

The table entitled Evidence of m6A function in liver pathologies should be revised by enlisting writers, readers, and erasers separately. Also, some of the text parts in that table should be explained in the manuscript text rather than in the table (the parts without appropriate writers, riders, and erasers).

In many of the studies referenced, multiple m6A RWE were studied. For example, one study that perturbed the expression of an m6A writer AND an m6A eraser so it would be difficult to separate the impact of each. We have written out the designation for each m6A modifier to make it easier for readers.

The authors should more carefully subdivide the manuscript text into appropriate subsections/paragraphs. For example, the text encompassed by lines 297 to 313 should be separated from the previous text.

We appreciate the Reviewer’s suggestion and have expanded upon this section to include a recent paper reporting increased m6A in CCl4-induced liver fibrosis in mice with increased m6A in NR1D1 being read by YTHDC1, reducing transcript stability.  The revised text is now lines 300-321.

Round 2

Reviewer 1 Report

The authors have addressed all my comments for this paper. The paper has been significantly improved after revising.  I recommend the editor accept this paper.